# Small & Medium-Sized Enterprises, Organizational Resilience Capacity and Flash Floods: Insights from a Literature Review

**Antonis Skouloudis** [1] , **Thomas Tsalis** [2], **Ioannis Nikolaou** [2,*], **Konstantinos Evangelinos** [1] **and Walter Leal Filho** [3,4]

1   Department of Environment, University of the Aegean, Xenia Building, 81132 Mytilini, Greece; skouloudis@env.aegean.gr (A.S.); kevag@aegean.gr (K.E.)
2   Department of Environmental Engineering, Democritus University of Thrace, Vasilissis Sofias 12, 67100 Xanthi, Greece; ttsalis@env.duth.gr
3   Research and Transfer Centre "Sustainability and Climate Change Management" (FTZ-NK), Hamburg University of Applied Sciences, Ulmenliet 20, 21033 Hamburg, Germany; w.leal@mmu.ac.uk
4   Department of Natural Sciences, Manchester Metropolitan University, Chester Street, Manchester M1 5GD, UK
*   Correspondence: inikol@env.duth.gr

**Abstract:** From a managerial standpoint, sustainability poses numerous challenges for the business community. One of the prominent concerns in the context of organizational sustainability is the impact of climate change and extreme weather events (EWEs), which create discontinuity and damages to business operations. In this respect, small and medium-sized enterprises (SMEs) are particularly vulnerable to EWEs, such as flash floods, having disastrous consequences to SMEs that tend to be ill-prepared. Taking into consideration that these negatives effects are also transferred into the local communities in which SMEs are located, it is crucial to create appropriate mechanisms that will enable these enterprises to build relevant capacities and acquire necessary resources in order to deal with relevant disruptive events. With this in mind, this paper attempts to delineate the emerging literature in relation to strategic approaches in dealing with high impact/low probability EWEs. With this analysis, we aim to provide insights for enhancing the robustness of SMEs against such natural hazards through effective resilience and adaptation strategies. The paper reveals that resilience to EWEs is indeed a multifaceted issue posing numerous challenges to SMEs. Taking into account their intrinsic characteristics, there is a need for a holistic management approach that will assist SMEs to safeguard their assets against extreme weather.

**Keywords:** climate change; resilience; extreme weather events (EWEs); small and medium enterprises (SMEs); floods

## 1. Introduction

In the new era of sustainability transitions defined by the launch of United Nations' 2030 Agenda for Sustainable Development, climate change adaptation sets key directions for formulating policies at global and national levels [1]. Particularly, sustainable development goal (SDG) 13 stipulates an array of targets that focus on improvements in climate-related resilience and adaptive capacity. In this context, scientific evidence supports that climate change impacts are pivotal challenges for sustainable development, threatening the balance of both natural and human systems [2–4]. Climate change is defined as the "*change in climate over time, whether due to natural variability or as a result of human activity*" [5] (p. 6), with the anthropogenic activities (causing excessive levels of greenhouse gas

emissions) recognized as having dramatic effects on the global climate system. Due to climate change, ecosystems and societies, all over the world, will be exposed to increasing risks and impacts [3,6–8].

Climate change is considered accountable for atmosphere and oceans warming, ice loss mass and sea-level rise [3,9]. It is also linked with extreme weather events (EWEs) such as flooding events, droughts, heat waves and storm surges, while it is anticipated to be a change in their frequency of occurrence, the duration and the magnitude of such events [6,7,10–13]. Current experience reveals that EWEs have increasing catastrophic consequences for local communities and society-at-large, creating discontinuities and adverse conditions due to asset and infrastructure damages [14–16].

Apart from the impacts on societies, EWEs pose a major risk to industries, threatening for-profit activities and may eventually force businesses to cease operations. From an organizational management standpoint, EWEs can be regarded as external shocks with high uncertainty [7,17–19]. For-profit entities are under continuous pressure to devise and maintain proper strategies and mechanisms that will allow them to effectively address EWEs impacts and, thus, reduce their relative vulnerability [4,20], i.e., the level of susceptibility to destructive impacts of climate variability and extreme weather [4,5] (p. 6). Vulnerability levels differ across business sectors and it is strongly associated with the relative exposure to EWEs of the area in which a business operates, as well as with the characteristics of each sector [5]. Agriculture, forestry, energy, oil and gas, insurance, tourism and construction industries are examples of business activities being particularly susceptible to EWE effects [5,8,11,15,21,22].

One of the most critical EWEs is flash flooding, which encapsulates abrupt and severe effects on businesses. As a result of heavy downpours and thunderstorms, such flooding events are expected to increase in absolute numbers, placing greater stress on organizations [13,23,24], which have to face a wide range of effects such as damage to assets and infrastructure, difficulties in daily operations, increased insurance premiums as well as impacts related to human capital [11,13,25].

Regardless the vulnerability level of firms and the severity of the direct (e.g., property damage) and indirect (e.g., insurance costs) impacts of flash floods in particular and EWEs in general, businesses have to be well-prepared to deal with such 'acute business interruptions' that lead to excessive discontinuities and increased repair costs [26] (p. 583). While it is difficult to predict the occurrence and the intensity of such events [7], businesses need to develop and implement agendas for action that will enable them to gain necessary resources and competencies in order to deal with flood risks. One critical notion in the context of business preparedness to cope with and overcome such events is the organizational resilience capacity [7,27,28]. Many definitions of organizational resilience have been set forth in an attempt to emphasize on diverse perspectives describing the ability of organizations to resist and recover, to adapt and anticipate low probability situations and high impact events [29,30]. With respect to EWEs, resilience capacity can be defined as "*the organizational capacity to absorb the impact and recover from the actual occurrence of an extreme weather event*" [10] (p. 2). It is a multidimensional construct reflecting the ability of an organization to experience a disruption without drastically affecting its normal operation or the capacity to bounce back from the negative impacts of an EWE and quickly recover (at least) to its original state [7,11,23,27,31,32]. Linnenluecke and Griffiths [33] present two fundamental dimensions of organizational resilience, namely "rapidity" and "impact of resistance," while the understanding of the vulnerability is a crucial factor that shapes the directions for improving organizational resilience [20]. By assessing their relative vulnerability, organizations are able to engage in capacity-building, which equips them to address the unpredictability and severity of EWE impacts [6,20,30]. The development of resilience capacity is a dynamic and continuous process through which organizations shape new capabilities and establish new routines as well as procedures that contribute to accomplishing various aspects of organizational resilience, such as the anticipation, extended coping and recovering range, along with increased adaptation potential over EWEs [7,10,29,30].

With projections of EWEs occurrence indicating that such unexpected natural hazards will be more frequent and severe, organizational resilience capacity should be regarded as an invaluable ability towards business continuity in order to reduce detrimental impacts of environmental perturbations

on their daily operations and production processes. Apart from direct benefits related to the ability to withstand external weather-related shocks, organizational resilience capacity is also an important business attribute in developing sustainable competitive advantages that endorse long-range planning and growth [27,29,30]. Hence, such essential advantages derived from building resilience could act as strong and meaningful incentives to motivate businesses to nurture and promote essential resilience-specific as well as sustainability-oriented capabilities and resources.

Additionally, conceptual underpinnings of organizational resilience to weather extremes set forth a new prospect for corporate environmental management and strategic planning, under the scope of the inadequacy of existing environmental management systems to address challenges and impacts linked with EWEs [6]. This is because environmental management approaches mainly focus on assisting a business to understand how their various operations and products/services affect environmental quality and how to implement effective policies, plans and programs to minimize negative environmental externalities. While this approach and point-of-view of environmental management frameworks is vital for organizational sustainability and businesses' contribution to sustainable development, it is insufficient in terms of elements and features that a business encounters from an outside-in perspective when they face climate or weather-related threats [6].

As adverse and intense impacts of EWEs, including flash flooding, are nowadays far from negligible, affecting societies and business systems worldwide, scholars started placing specific attention on how small and medium-sized enterprises (SMEs) can be better prepared to deal with such environmental perturbations and, ultimately, what drives their ability to configure appropriate responses and build resilience [2,13,23,34–42]. Crucially, the impacts of flash floods (among other natural hazards) on SMEs can be greater and more severe compared to their larger business entities [13,15,40]. SMEs are extremely vulnerable to flooding [25] and have been characterized by low level of resilience and insufficient preparedness to confront such events [23,40]. Various factors have been identified as explanatory parameters [39] with limitations in financial, managerial and human resources as primary ones [2,15,23,37,38]

In this respect, it is of critical importance to examine the wide spectrum of factors which facilitate or discourage SMEs to develop their resilience capacity due to the fact that the impacts of EWEs on SMEs could also bring significant problems at local, regional and/or national levels (for instance, supply chains experiencing long-term interruptions or ceasing to function). This is owing to the crucial role of the SMEs in the local societies as job providers, and another explanation is that SMEs comprise the vast majority of businesses operating both in developed and developing countries [15,23,25,41–43]. Therefore, the great impacts of SMEs on the economic development, at all levels, clearly shows the necessity for effective tools for preparing them for EWEs.

## 2. Theoretical Background

The occurrence of EWEs can result in extremely negative environmental, social and economic impacts. Bergmann et al. [44] explored the effects of the different types of EWEs (e.g., cold waves, severe thunderstorms and flash floods) on various organizational operations and aspects, e.g., procurement operations, marketing and services, logistics and human resources. In this context, Linnenluecke et al. [10] suggested a critical and instructive classification for EWEs in three groups: simple extremes (local phenomena based on clear variables), complex extremes (local phenomena relied on a variety of variables) and unique extremes (global phenomena). The negative impacts of EWEs differ among various economic and social actors such as public authorities and local communities [45]. Particularly, previous studies reveal that EWEs can bring adverse effects on construction industries [46,47] and the tourism sector with shorter seasons, transport disturbance, less security and loss of revenue [21] as some of the critical impacts.

Previous studies also indicate that the level of influence of EWEs varies across firms [34,35]. As mentioned above, firm size has been identified as key factor explaining the variation in vulnerability to EWEs. The impacts of EWEs are more disastrous on SMEs than on larger firms [33,48] and SMEs

encounter many obstacles in their efforts to face extreme weather. Such barriers are mainly associated with the lack of financial capital, inadequate know-how as well as limitations in technological competencies and skilled human resources [49]. Runyan [50] pointed out that due to the limited resources SMEs are ill-prepared to achieve a quick recovery from EWEs. However, a contrary view holds that some of the SMEs' features may offer them an advantage in order to cope with EWE impacts [51]: low level of bureaucratic processes, quick decision-making or effective internal communication and routines for an immediate implementation of strategies [49].

In the field of SMEs vis-à-vis EWEs, there is an urgent need to devise and disseminate effective and efficient ways to assist SMEs in dealing with the underlying negative impacts of EWEs and ensure business continuity. Against this background, numerous concepts (i.e., business resilience, business vulnerability, business adaptation, business continuity, organizational coping strategies, risk management and natural hazards crisis management) have been coined to outline management practices necessary for firms to confront EWEs as well as management tools developed to assist SMEs to withstand and overcome these types of environmental change. For instance, Wedawatta and Ingirige [52] proposed a management system approach in order for SMEs pertaining to the construction industry to effectively cope with EWE damages through a triangulation of vulnerabilities (e.g., size of SMEs, location of projects, firm specialization), coping strategies (general risk management, coping strategy at business level) and coping adaptation (e.g., previous experience with EWEs, financial resources). In a similar vein, Bostick et al. [53] suggested a stakeholder-based multicriteria model to assist firms in decision-making concerning their resilience status, which consists of five stages: moderated discussion (e.g., resilience, system domain), stakeholder input, decision-maker input, model, output, and reassessment. Likewise, Shashi et al. [54] proposed a conceptual model to classify the current literature in organizational resilience regarding supply chain management. Specifically, their contribution examines the business resilience strategy in the context of the supply chain which can be divided into three overarching domains: anticipation (e.g., capability, distribution management and strategy formation, planning and design, and properties), resistance (e.g., supply chain reengineering, collaboration, agility, and supply chain risk management culture) and recovery-response actions (e.g., recovery preparation, long-term impacts). Haraguchi et al. [55] set forth a business continuity management model based on public-private partnerships to face EWEs. According to this model, business resilience is classified into four levels: firm level resilience, supply chain resilience, public-private level resilience, and societal resilience. Linnenluecke et al. [10] pointed out a framework to strengthen business resilience which comprises of three parts. The first one includes the anticipatory adaptation strategy, examining the previous experience of business regarding EWEs; the second pertains to organizational capabilities developing a management algorithm to examine sense-making of disaster, sensitivity, disaster response and reconstruction; while the third part refers to procedures for future adaptation strategies addressing future organizational capabilities in confronting EWEs. In a similar vein, Linnenluecke, et al. [11] proposed a relocation model for firms to deal with EWEs, relying on environmental sensitivity factors, feasibility of strategy implementation along with the relocation costs.

SMEs need to develop and deploy strategies in order to successfully recover and maintain their organizational viability after an abrupt, unexpected and disastrous flooding event. From a theoretical standpoint, such business resilience strategies can be explained through various conceptual frameworks and analytical lenses (Table 1). All these theoretical frameworks have been utilized to disaggregate the different approaches and explain firms' responses to the challenges arising from sustainable development under the scope of climate and weather-related hazards. A common ground for the development of these theories is that they recognize that the mere focus on financial goals is inadequate to guide firms to success. Environmental and social parameters should be integrated into corporate strategy in order for firms to thrive in a complex and turbulent environment.

**Table 1.** Theoretical background of organizational resilience to extreme weather.

| Theoretical Lens | Key Points | Authors |
|---|---|---|
| Organizational theory | Organizations' ability to respond to EWEs as well as to adapt their processes in order to make new responses. | [7,31,33,35] |
| Institutional theory | An organizational adaptive capability is associated not only with their internal capabilities, but also with the external environment (e.g., social, political, and economic environment). | [6,56,57] |
| Systems theory | Business and external environment are interrelated variables. | [45,58,59] |
| Resource dependence theory | Business operation dependence on natural and ecological resources. | [44,60,61] |

A well-established theory to explain the reaction of firms to flash floods and other EWEs is the organizational theory and behavior. Under this theoretical lens, there are two fundamental approaches, namely reactive and proactive responses to external stimuli. While the former focuses on the ability of a firm (organization) to overcome unexpected events, the latter examines not only the organizational capabilities to deal with extreme events but also how these capabilities can allow firms to identify or create new opportunities in a timely manner [62]. An indicative example of the proactive approach can be found in the work of Linnenluecke and Griffiths [33], who suggest the need for making better links between organizational resilience and adaptive response strategies in order for organizations to successfully withstand, absorb and eventually recover from the occurrence of unexpected weather extremes such as flash flooding.

Institutional theory has also been employed to explain business resilience strategies and the level of resilience capacity demonstrated. According to this perspective, for-profit activities and the adaptive strategies for coping with EWEs (as abrupt and unexpected changes) should not only be associated with the internal organizational capabilities but with the enabling conditions provided by the institutional environment as well [56]. Winn et al. [6] suggested that institutional theory offers an extremely valuable and fruitful context to analyze how organizational adaptation processes are adopted, shaped and endorsed within the enterprise. In a similar vein, focusing on Scandinavian business systems, Wejs et al. [57] identified an array of institutional factors affecting companies through both anticipatory and mandatory actions in order to implement climate change adaptation strategies.

Systems theory has also been proposed as a theoretical lens to shed light on business vulnerability, adaptive capacity and resilience potential [58]. In line with systems thinking, organizations and their external environment consist of a complex and dynamic system where there are strong interrelationships between its components. Through systems theory and system dynamics (SD) modeling tools, Nikolaou et al. [45] analyzed potential impacts from physical risks, such as droughts and floods, on business operations. The core findings of this model indicate the strong relationship between physical risks and financial performance of business entities. It is also suggested that floods (amplified by long-term global climate change) threat business continuity through discontinuities in the supply chain and daily operations. Managers need to overcome such problems through new investments in equipment and recovery measures. Similarly, Tsalis and Nikolaou [59] proposed a system dynamics model in order to manage risks faced by firm due to climate change. Their model identifies a significant influence of climate change risks on business economic performance. Conceptual models such as the above attempt to shed light on the key relationships of flash flooding effects on business performance through a systems theory lens.

Organizational responses to EWEs have also been studied through a natural resource dependence theoretical perspective where the ecological resilience paradigm is introduced into organizations' strategic management. The underpinnings of this theory in relation to business planning posit that every entity (human or business) depends on the ecosystem (and its biophysical processes) as it needs an array of natural resources to survive and (eventually) thrive. In this context, Bergmann et al. [44] employed resource dependence theory to explain how EWEs affect the financial performance of businesses. Similarly, based on resource dependence and institutional theories, Tashman and Rivera [61] pointed out

a critical relationship between the status of the US ski resort industry and climate change implications and point out the notion of ecological uncertainty as a supporting argument of the difficulty of businesses to gain access to vital natural resources. In this respect, in order to overcome ecological uncertainty, it is suggested that businesses should adopt "natural-resource-intensive practices" in order to moderate and overcome its negative impacts [61].

Nevertheless, fundamental questions still remain on how businesses can overcome the effects of EWEs such as flash floods as well as whether the organizational capabilities are sufficient to overcome the negative impacts of such environmental perturbations, or why firms should cooperate with key social constituents/stakeholders to increase their preparedness against natural hazards. The emerging literature on the specific 'business and the environment' domain does offer insightful theoretical explanations on why businesses engage in (proactive or reactive) efforts in confronting EWEs, and also places emphasis on how SMEs can sufficiently adapt to extreme weather, minimize the impacts of such disturbances and boost their performance in an uncertain environment.

In this respect, in order to provide a general outline of such practices, a novel (rudimentary) framework is suggested, classifying them into three strategic layers denoted as micro-, meso- and macro-level (Figure 1). The micro-level refers to capabilities of SMEs which are critical for the effective planning, mitigation, adaption and recovering processes in relation to EWEs and their consequent impacts [13,63]. It can be based on organizational, resource- and knowledge-based theories where businesses face the negative impacts of a highly unpredictable environment through their capabilities and resources and manage to return in the initial state by creating new opportunities for sustainable competitive advantage (organizational-based theory). Such theoretical lens can be helpful to businesses that have sufficient resources (e.g., financial capital and skilled human resources), previous experience with EWEs (e.g., existing capabilities, knowledge-creating routines and adaptive procedures) and/or demonstrate a low level of vulnerability with negligible impacts of EWEs on their operations.

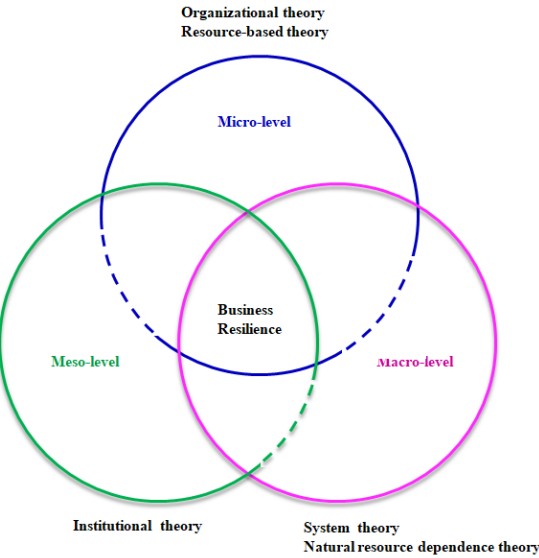

**Figure 1.** Organizational resilience to extreme weather events (EWEs)—a general framework of theoretical perspectives.

Natural resource-based theory, knowledge-based theory and intellectual-based theory offer a concrete context to explain how organizational responsibility and environmental management practices provide incentives and motivate enterprises towards better performance and promoting long-term businesses growth [64,65]. The basic principles of such theories rely on capabilities, skills, resources and competencies of businesses to face modern challenges. Crucially, these business attributes (e.g., technological competency, design procedures, procurement strategies, production processes, distribution channels and service capabilities) can 'shield' the organization from external risks. In this

logic, businesses with specific capabilities and resources as well as intellectual capital creating knowledge (tacit, social complex and rare) can successfully confront environmental perturbations and change, such as flash floods and other EWEs [66].

The meso-level implies that merely relying on business capabilities is not enough to successfully bounce back from EWEs. Some disturbance in business operations may arise from problems caused in the supply chain and in other business partners or regions. Actually, EWEs may have significant impacts on the supply chain that can indirectly affect business operations. Wedawatta et al. [67] identified that over 50% of the problems stemming from EWEs in the UK SME construction industry are associated with supply chain issues (e.g., suppliers' disruptions, loss of energy and water supply). Some significant problems in the supply chain derived from EWEs affecting business operations can also be delays on scheduled procurements and logistics disruptions [68]. Businesses can overcome such issues through participatory activities with governmental bodies, business chambers/associations and supply chain managers in order to promote knowledge sharing among key actors [68].

It is significant to point out that many reactions of businesses on climate change problems are strongly associated with institutional pressures and could be explained through the institutional theory lens. Institutional theory posits the many types of external pressures that stimulate enterprises to adopt strategies to address environmental and climate change problems. For instance, Escobar and Vredenburg [69] pointed out the three forms of isomorphism described by neo-institutional theory (coercive isomorphism, normative isomorphism and mimetic isomorphism) in explaining sustainability transitions in firms. The first two types reflecting aspects of the regulatory regime can affect decisions of businesses regarding climate change and weather extremes impacts. The third form explains business climate change adaptation and resilience building behavior as a mimetic process driven by peer firms. This (mimetic) effect can be placed in the second (meso-) level while the first two forms of isomorphism in the third (macro-) level (indicated by the dashed green line in the figure). It is worth noting that cooperation of businesses at the meso-level could be so explained from a systems theory perspective. Several scholars suggest that cooperation of businesses in an industrial ecology context plays a critical role in resilience capacity building against environmental change [70,71]. This viewpoint also lends support to the theoretical connection between institutional theory and systems theory to further clarify how business participatory and multi-stakeholder actions can be a meaningful planning endeavor to address EWEs.

The macro-level encapsulates collaborative activities of businesses not only with governmental bodies and other businesses but also with local communities and third sector organizations (NGOs) in order to overcome problems linked with the occurrence of EWEs. Systems theory explains the necessity of business cooperation with various social constituents and economic actors in order to promote resilience and ensure continuity. To build a robust level of resilience capacity against flash flooding and other EWEs, enterprises should engage and cooperate with other societal agents and local community members. In this respect, Wyss et al. [19] suggested that the cooperation of such various individual agents, due to relative independencies and mutual interests, is a necessary condition for resilience and adaptation processes in the tourism and hospitality sector as the support of governmental authorities as well as media, NGOs and local community is deemed to be vital. This approach can be explained through the systems theory and the natural resource dependency theoretical perspectives.

## 3. An Overview of Empirical Studies on SMEs Resilience to Weather Extremes

Over the past decade an emerging wave of empirical studies around the world have sought to explore how SMEs are affected by EWEs and flooding specifically, their coping range of strategies as well as inhibitory factors to adaptation and organizational resilience (see Table 2). Such research endeavors attempt to interpret the underlying threats and opportunities stemming from resilience capacity (or the lack thereof) SMEs demonstrate.

Hermann and Guenther [72] assessed SMEs' barriers to adopting climate change adaptation strategies in a large city in Germany. Following a questionnaire survey method, a barrier scale

was developed allowing causal explanations for the occurrence of barriers and how they can be managed and addressed. Likewise, Halkos et al. [35] and Halkos and Skouloudis [36] investigated resilience barriers to EWEs and flooding among Greek SMEs through structural equation modeling and quantile regression analysis, allowing for fruitful insights and essential, context-specific evidence for practitioners and policy-makers respectively.

Karman [73] investigated individual, organizational, community-specific and extreme-related factors affecting the resilience mechanisms applied by business entities from 20 European countries. Aiming to provide a better understanding of business resilience to weather extremes, the study sheds light on the relative frequency particular mechanisms (including disposition and administration of resources, self-organization, intra-organizational communication, damage assessment, review of previous events and the acquisition of external information) are applied in and verifies determinants of their employment.

Mullins and Soetanto [74] focused on the relative importance ethnic differences and demographic factors have in the disaster management field linked to flooding in Birmingham (UK) communities. By employing a quantitative approach in data collection, they found three levels of resilience and an association of those with different ethnic groups, as well as that ethnic differences consistently exist within the perceptions of business groups within the study's communities that have recent experience of flooding, but not in a community without recent flood experience.

Wedawatta et al. [68] employed a mixed methods research design to elicit data on how construction SMEs located in the Greater London area respond to EWE risks and stress that the coping strategies implemented leave much to be desired. In this respect, the authors stress the need for better integration of EWE occurrence into initial project planning stages through better risk assessment models as well as more accurate EWE prediction data. Similarly, Ingirige et al. [75] examined impacts of flooding on SMEs in Cockermouth (Cumbria, UK) using a mixed method of interviews with experts having long-standing experience in advising SMEs on post-flood reinstatement along with a questionnaire survey to 48 SME owners/managers. The findings of the study provide fruitful and actionable insights on chartered surveyors' capacity-building in the field of SME adaptation to flood risk under the scope of reliable and valid advice on property-level flood protection measures.

Kuruppu et al. [76] conducted a mixed method approach involving a set of semi structured interviews, case studies and a workshop to examine underlying factors and processes shaping the adaptive capacity and resilience potential of Australian SMEs to climate change and weather extremes. The study highlights the critical importance that contextual processes encapsulate in enhancing the adaptive capacity of SMEs, and Kuruppu et al. [76] pointed out that contextual processes had been largely overlooked in formal programs aiming to build business resilience, being primarily reactive and focusing on recovery during and after disasters, rather than on anticipatory prevention and preparedness.

Wedawatta and Ingirige [13] conducted a number of short case studies among UK SMEs to identify responses to flood risk as well as measures undertaken to address impacts. The authors observed that, following a post-flood situation, SMEs tend to implement diverse property-level protection measures and generic business continuity/risk management practices, according to individual requirements, with the overarching aim of achieving a desired status of flood protection. Ingirige and Russell [77] also employed a case study analysis in seven SMEs in Braunton (North Devon, UK), offering valuable evidence across a range of enterprises and highlighting innovative approaches to flood impact mitigation. Aiming to contribute to behavioral changes, the report finds that interviewed SMEs became 'experts by experience' on those resilience measures they implemented and highlights the enthusiasm among the SME community for sharing and enhancing their capacities further.

**Table 2.** Empirical studies assessing small and medium-sized enterprises' (SMEs) responses to EWEs/flooding stimuli.

| Year | Author(s) | Journal/Outlet | Country(-ies) | Method(s) | Analytical Lens |
|---|---|---|---|---|---|
| 2011 | Wedawatta, Ingirige, Jones and Proverbs | Structural Survey | United Kingdom | Mixed methods | Micro-level coping strategies |
| 2012 | Wedawatta and Ingirige | Disaster Prevention & Management | United Kingdom | Semi-structured interviews | Micro-level coping strategies |
| 2012 | Ingirige, Proverbs and Wedawatta | RICS Education Trust | United Kingdom | Mixed methods | Organizational/micro-level; resource dependency; institutional capacities |
| 2013 | Wilk, Andersson and Warburton | Regional Environmental Change | South Africa | Interviews | Organizational/micro-level; institutional capacities |
| 2013 | Mullins and Soetanto | Disaster Prevention & Management | United Kingdom | Questionnaire | Informal institutions (cultural norms) |
| 2013 | Kuruppu, Murta, Mukheibir, Chong and Brennan | National Climate Change Adaptation Research Facility | Australia | Mixed methods | Organizational and meso- level; institutional capacities |
| 2015 | Ingirige and Russell | UK Climate Impacts Programme, University of Oxford | United Kingdom | Interviews | Micro-level coping strategies; resource dependency and institutional capacities |
| 2015 | Li, Coates, McGuinness and Johnson | International Conference on Flood resilience Zurich, Switzerland, 13–14 January | United Kingdom | Semi-structured interviews & agent-based modelling | System dynamics; micro- and macro-level interactions |
| 2015 | Ballesteros and Domingo | Philippine Institute for Development Studies | Philippines | Secondary data analysis | Organizational responses and macro-level/institutional support |
| 2016 | Li and Coates | International Journal of Design Nature & Ecodynamics | United Kingdom | Semi-structured interviews & agent-based modelling | System dynamics; micro- and macro-level interactions |
| 2016 | Pathak and Ahmad | International Journal of Disaster Risk Reduction | Thailand | Mixed methods | Micro-level responses and institutional capacities; macro-level supporting mechanisms |
| 2017 | Hermann and Guenther | Journal of Cleaner Production | Germany | Questionnaire | Organizational capacities & resource dependence |
| 2017 | Mark and Thomalla | Natural Hazards | Thailand | Mixed methods | Micro-level responses and system dynamics/macro-level support |
| 2017 | Kato and Charoenrat | International Journal of Disaster Risk Reduction | Thailand | Questionnaire | Organizational/micro-level; institutional capacities |
| 2018 | Samantha | Procedia Engineering | Sri Lanka | Semi-structured interviews | Organizational/micro level |
| 2018 | Alharbi and Coates | WIT Transactions on The Built Environment | United Kingdom | Semi-structured interviews & agent-based modelling | System dynamics; micro-level responses & institutional capacities |
| 2018 | Halkos, Skouloudis, Malesios and Evangelinos | Business Strategy & the Environment | Greece | Questionnaire | Organizational/micro-level |
| 2018 | Crick, Eskander, Fankhauser and Diop | World Development | Kenya, Senegal | Questionnaire | Organizational/micro-level |
| 2020 | Halkos and Skouloudis | Climate and Development | Greece | Questionnaire | Organizational/micro-level |
| 2020 | Karman | Business Strategy & the Environment | 20 European countries | Questionnaire | Micro- & meso-level intreactions; system dymanics |
| 2020 | Coates, Alharbi, Li, Ahilan and Wright | Philosophical Transactions of the Royal Society A | United Kingdom | Semi-structured interviews & agent-based modelling | System dynamics; micro- and macro-level interactions |

Utilizing an agent-based simulation model to assist UK SMEs facing flood disruptions Li et al. [38] and Li and Coates [78] offered evidence towards the development of effective response strategies which SMEs can employ to reduce the flood impacts, better assess the level of continuity of operations and, ultimately, increase their resilience. In a similar vein, Alharbi and Coates [79] focused on UK manufacturing SMEs in Sheffield (UK) and model SMEs' behaviors that can be enacted pre- and post-flood and shed light on the influence of different types of insurance coverage and financial status on the response and recovery from different levels of flooding, in attempt to indicate the influence of combinations of these attributes on SME recovery. More recently, Coates et al. [23] provided findings of an application of a similar computational modeling and simulation approach to evaluate SMEs' operational resilience to extreme floods based on combinations of structural and procedural mitigation measures that may be implemented to improve SMEs resistance to flooding and ensure business continuity. Using the major flood event of 2007 in Tewkesbury (UK) as case study, the assessment enables an evaluation of operational resilience of manufacturing SMEs in terms of the relative effectiveness of flood mitigation measures and stresses that structural mitigation measures are more effective compared to procedural ones.

Kato and Charoenrat [80] investigated business continuity management practices employed by Thai SMEs in order to highlight underlying assistance needs. Analyzing questionnaire-based data gathered from SME managers, the study confirms the increased disaster experience of Thai SME and points out inadequate levels of preparedness towards business continuity planning allowing to suggest the critical importance of extending support to SMEs in disaster-prone areas. Pathak and Ahmad [41] employed a mixed methods approach in order to examine flood recovery capacities adopted by SMEs affected by flooding in the Pathumthani province (Thailand). Focusing on manufacturing SMEs the study provides fruitful evidence of coping strategies and in ascertaining the impacts of flood disasters in the area. In a similar vein, focusing in a flood-prone area of the Bangkok Metropolitan Region, Mark and Thomalla [42] examined SME responses and recovery from the 2011 Bangkok floods and measures taken to reduce the vulnerability to future floods. By conducting in-depth key informant interviews and a questionnaire survey with SME owners, the authors shed light on how (and the extent to which) SMEs were affected by the 2011 Bangkok floods and actions by SMEs and governmental bodies respectively in order to reduce vulnerability to future flooding. The study concludes that socioeconomic factors interacted with the 2011 flood to negatively affect SMEs, as well as that key political economy drivers of vulnerability of SMEs are far from addressed.

Crick et al. [81] reported on the extent to which micro enterprises and SMEs in Senegal and Kenya are adapting to climate risks. Drawing from findings derived from a questionnaire survey on SMEs in semi-arid regions in these countries, the assessment estimates the maturity of adaptation measures in place and attempts to distinguish between sustainable and unsustainable adaptation. The study encapsulates meaningful implications for policy interventions in building resilience to future climate risks by indicating a number of factors affecting the level of organizational adaptation to current climate variability: availability of financial resources, general business support, access to information technology and adaptation assistance.

Focusing on SME sector in the Philippines, Ballesteros and Domingo [82] set forth strategic recommendations for local and national policy design in order to embed disaster risk reduction and management into the SME planning, and stressed the key role of the regional economic forum of Asia-Pacific Economic Cooperation (APEC) for endorsing the resilience of member-countries' SMEs towards natural hazards. Similarly, Samantha [43] conducted semi-structured interviews with micro and SME owners regarding the adverse impacts of flooding in Sri Lanka and provided recommendations on strategic multi-stakeholder policies to disaster risk reduction and disaster coping mechanisms into the respective business sectors. The qualitative data allowed to outline organizational experiences on various aspects of damage, rehabilitation and re-establishment, and indicated specific vulnerability points within the enterprise in terms of capital, labor, logistic and market impacts.

Wilk et al. [83] conducted interviews with commercial and small-scale farmers in South Africa in an attempt to frame challenges and adaptive strategies to address climate-related stressors and EWEs. The analysis suggests that small-scale farmers tend to be more vulnerable due to factors such as the limited access to finance as well as to agricultural techniques for water and soil conservation along with the high input costs of improved seed varieties. In contrast, commercial adaptation strategies were primarily hindered by the vague governmental directives towards sustainable agriculture and the climate-proofing of the agricultural production. Being part of a larger participatory (climate) adaptation planning project with local stakeholder groups, the study concludes that knowledge transfer within and across farming communities, clearer governmental directives and targeted, locally adapted finance programs should be the best way forward.

Studies such as the above offer multiple actionable insights and provide implications to SME management and policy-design in achieving a climate-proof and EWE-resilient SMEs sector. Nevertheless, reflecting on the available literature, much work needs to be done to provide the enabling conditions for SMEs to better prepare and successfully overcome such environmental perturbations.

## 4. Conclusions and Implications for Future Research

Undoubtedly, changes in weather patterns due to climate change and the increase of EWEs in absolute numbers create a new reality for the business community. Special attention should be devoted to flash flooding, which emerges as one of the most critical EWEs with abrupt and disastrous consequences for business and society [38,41,84,85]. Given that SMEs are particularly vulnerable to EWEs, lacking adequate resources and managerial skills to minimize the negative impacts and successfully recover from such disruptions [15,23,43,81,86], it is crucial to assess the wide range of factors associated with the internal and external business environment in order for SMEs to become better-prepared against flooding and its damaging effects. Supporting arguments for this claim can also be found in previous studies on flood impacts, which indicate that such events can be a defining moment in SME operation, causing numerous severe damages and, in a worst-case scenario, forcing them to cease operations [25,41,42,87].

Outlining the relevant literature, a key finding is that there is not a widely applicable management approach for addressing challenges accruing from flash flooding events. Although there is a growing body of research on this field, the majority of previous studies have employed questionnaire-based surveys or semi-structured interviews in order to elicit various factors and approaches adopted by firms and associated with their resilience capacity, vulnerability to weather extremes and their preparation level for future flooding extremes. These studies mainly document previous experience or analyze the mechanisms and response (ex-post) strategies developed by firms in order to increase their resilience capacity. Undoubtedly, such information is necessary for understanding the context in which enterprises operate in relation to EWEs but it is insufficient in guiding them to opt for the appropriate measures that will reduce their vulnerability to future flooding events. This is because most of such studies fall short in proposing scalable tools and s.m.a.r.t. targets (specific, measurable, achievable, realistic, and timely) that will definitely help SMEs to assess the effectiveness of various flood protection measures, taking to consideration their intrinsic characteristics.

Research on SMEs' resilience capacity to EWEs, and flash floods in particular, leaves much to be desired and should be advanced on its own merits beyond mere rhetoric and anecdotal evidence or particularly fragmented data. With this in mind, there are some fruitful directions for future research concerning the preparedness of SMEs to EWE threats in order take advantage of essential benefits accruing from bouncing back and eventually thriving after such events. Flash flooding events encapsulate multiple and diverse impacts on business, which can be closely interrelated, complicating organizational efforts to build efficient mechanisms to deal with such natural hazards. This is evident from the various approaches and criteria proposed to categorize floods impacts on firms. Apart from direct impacts (such as damages to business premises and equipment, injuries as well as losses of raw materials and stock), there are indirect impacts that can create serious

obstacles to business continuity, i.e., problems associated with the supply chain, human resources and logistics [13,25,43,88,89]. It is also worth mentioning that firms that have not been physically affected by floods can also experience indirect impacts from these environmental perturbations [25]. The temporal dimension is another aspect employed to classify impacts of floods on firms into long- and short-term impacts [25,43]. According to Wedawatta and Ingirige [13], damages to capital assets are indicative examples of short-term impacts, while low income and high insurance premiums pertain to long-term impacts among others. Additionally, flash flood impacts can be examined in relation to aspects of business operation affected. In this respect, Metcalf et al. [90] propose a list of climate change impacts namely, markets, logistics, premises, people, procedure and finance while Ballesteros and Domingo [82] define four aspects of business operation affected by natural disasters: capital, logistics, labor and market/buyers (see also [43]). In light of the above, flash floods contribute to a dynamic and complex environment in which firms have to develop increased resilience and adaptation capacities. It is essential for SMEs to gain a full understanding and appraisal of all the dynamic multidirectional interactions between flooding impacts and business operation, time lags which exist in these interactions and their effects on organizational performance over time. Considering the limited resources of SMEs, SD can be a promising approach in facilitating SMEs to respond to management challenges arising from floods. Both qualitative and quantitative tools of SD may give room to SMEs to assess how a flash flood can affect various business aspects and to evaluate the outcomes of alternative strategic scenarios (e.g., through quantitative simulation models) or perform a what-if analysis testing of short- and long-term implications from flooding [59,90–92]. Such feedback can be a valuable input for shaping strategies and developing mechanisms for adequate protection from floods. Thus, future empirical studies could emphasize on the SD approach and its application in facilitating SMEs to enhance their resilience capacity to flash floods and other EWEs.

Furthermore, a comprehensive analysis of past flooding events and the assessment of their impacts on SMEs can be a meaningful approach in advancing our understanding of how various internal and external measures affect SMEs' level of resilience capacity [40,43]. By examining SMEs that have previous experience with flash floods, in-depth knowledge can be obtained on the effectiveness of strategies and measures employed in order to reduce impacts of and contribute to the recovery process. While several recent studies have sought to analyze impacts and factors associated with the recovery from floods and other EWEs [40,41,43,93,94], more empirical research is required in order to gain a better understanding of particular measures and actions that facilitate SMEs to robustly address short- and long-term flood impacts. Such knowledge, which can also be gained through the application of composite firm-level indicators assessing organizational, behavioral and contextual factors of the resilience capacity level, can serve as a basis for developing sets of actionable guidelines of good practices that may be adjusted to individual needs and adopted by SMEs in order to strengthen their resilience capacity. This can be achieved in collaboration with critical stakeholders in order to plan and implement agendas for action that will enhance the resilience at a community or regional level [90,95].

Lastly, in line with the above research recommendations, it is essential to consider and examine in detail the role of the particular internal characteristics that distinguish SMEs from other firms and pose barriers in their efforts to manage challenges and tensions linked to (previously unforeseen) disruptive events such as flash floods [23,49,96]. For an SME-specific flash flood management system to be robust and effective, additional research shedding light on and allowing to overcome these barriers is essential. Research endeavors focusing on these barriers can contribute to transforming such obstacles into new opportunities for securing performance and continuity while minimizing negative impacts and bottlenecks associated with flash floods among other natural hazards.

**Author Contributions:** Original idea, A.S.; conceptualization, A.S., T.T., W.L.F. and I.N.; resources, A.S., K.E. and I.N.; literature review, T.T., A.S., I.N. and K.E.; writing—original draft preparation, T.T., I.N. and A.S.; writing—review and editing, T.T., A.S., I.N. and K.E.; discussion, T.T., K.E., and A.S.; supervision, K.E. and A.S.; project administration, A.S. and K.E.; funding acquisition, A.S. All authors have read and agreed to the published version of the manuscript.

**Funding:** The research work was supported by the Hellenic Foundation for Research and Innovation (H.F.R.I.) under the "First Call for H.F.R.I. Research Projects to support Faculty members and Researchers and the procurement of high-cost research equipment grant" (Project Number: HFRI-FM17-1844).

**Conflicts of Interest:** The authors declare no conflict of interest.

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
