# Peer review of "Small & Medium-Sized Enterprises, Organizational Resilience Capacity and Flash Floods: Insights from a Literature Review"

_sustainability, doi:10.3390/su12187437_

Round 1
Reviewer 1 Report
Abstract:
Minor grammatical check still needed. First sentence, needs “a” so “From a managerial..”. Also next sentence should be “are the impacts of climate change…”. The rest of abstract looks good.
Introduction and Theoretical Background:
Throughout narrative, error in references IPPC should be “IPCC…”. Page 2 line 76, should be “probability”.
Page 5, title Table 1. Font needs to be larger. Also, please don’t right justify the text in Table.
Page 6. Lines 223 should be “consist of a…” and line 228 “…that floods as a result…”
Page 7: lines 298 and 299 – Meso not messo.
An Overview of Empirical Studies:
Well done. Table 2 similar to Table 1 needs larger font.
Conclusions and Implications for Future Research:
Page 12: insert “the” in line 441: “…reality for the business community.” Begin with “Special..” not “A special.”
Page 12: line 478: What is “SD” and where is it defined in the paper?
In summary, a good overview of the literature relating to SMEs and EWEs, with particular attention to flooding, and sensible recommendations in moving forward with future research. After a minor grammatical check, primary suggestion would be to add a summary table of lit review of each reviewed paper’s primary objective, or add a column to Table 2 explaining what the research pertained to (e.g., organizational, institutional systems, etc.). Although explained in the narrative, a table with specialty areas would help readers to better discern the topics and approaches each of the lit reviewed authors examined and their particular area of research and methodology.
Author Response
Reviewer #1
Abstract: Minor grammatical check still needed. First sentence, needs “a” so “From a managerial..”. Also next sentence should be “are the impacts of climate change…”. The rest of abstract looks good.
- Response: The abstract was checked and suggested corrections are incorporated in the text.
<><><>
Introduction and Theoretical Background:
Throughout narrative, error in references IPPC should be “IPCC…”. Page 2 line 76, should be “probability”.
Page 5, title Table 1. Font needs to be larger. Also, please don’t right justify the text in Table.
Page 6. Lines 223 should be “consist of a…” and line 228 “…that floods as a result…”
Page 7: lines 298 and 299 – Meso not messo.
- Response: These minor grammar/syntax mistakes are now corrected in the revised manuscript; Table 1 will be modified accordingly by the editorial office if/when the submission is accepted and during the proofreading stage.
<><><>
An Overview of Empirical Studies:
Well done. Table 2 similar to Table 1 needs larger font.
- Response: Table 1 will be modified accordingly by the editorial office if/when the submission is accepted and during the proofreading stage.
<><><>
Conclusions and Implications for Future Research:
Page 12: insert “the” in line 441: “…reality for the business community.” Begin with “Special..” not “A special.”
- Response: These minor grammar/syntax mistakes are now corrected in the revised manuscript.
Page 12: line 478: What is “SD” and where is it defined in the paper?
- Response: SD refers to System Dynamics. The concept is first introduced in the manuscript in p.6, lines 224-225: ‘Through systems theory and system dynamics (SD) modelling tools,…’.
<><><>
After a minor grammatical check, primary suggestion would be to add a summary table of lit review of each reviewed paper’s primary objective, or add a column to Table 2 explaining what the research pertained to (e.g., organizational, institutional systems, etc.). Although explained in the narrative, a table with specialty areas would help readers to better discern the topics and approaches each of the lit reviewed authors examined and their particular area of research and methodology.
- Response: A column was added in Table 2 in order to indicate the core analytical lens(es) employed in each of the empirical studies we included this section.
<><><>
Reviewer 2 Report
This is a well-structured lit. review of the impact on SMEs by EWEs which includes a theoretical perspective as well as a review of multiple global examples. As such the paper is a welcome addition to the literature and I have no significant concerns with the material. The paper concludes, unsurprisingly, that this is a complex issue that requires a holistic management approach. It is a little disappointing that there is no simple solution but then that would be rather idealistic. I see no reason for the paper not to be published subject to one substantial change and some minor revisions.
The concluding section (4.) should reflect more on what the underlying reasons for the current lack of a holistic approach are rather than just revisit several key papers. A much stronger critique and way forward would be very helpful.
Minor comments (refers to line numbers)
- 46 - 'even their mere survival' is a bit dramatic - businesses always come and go - and they dont have a right to exist. A clearer position by the authors on what they see as central to sustainability (the economy?, people? environment?)
- 59 - 'profound'? - correct term?
- 140-141 - first sentence seems redundant
- 164 - 'serious and strong'? - says who?, why?
- Table 1 - reduce font size? Table 2012?
- Table 1 etc. what is the linking lens between these theories and which school of philosophy do they all come from? This needs teasing out. What is the position held by the authors and how are they viewing the SME world?
- 267 - remove 'actually'
- 293 - ()?
- 298 - messo / meso?
- 438 - 'excerpt' ? Table 2 - is this really necessary - not clear what it adds other than a list of pubs.
- 560/645/681/782 - font for the dod?
- 583 - weak reference - which jurisdiction etc?
Author Response
Reviewer #2
The concluding section (4.) should reflect more on what the underlying reasons for the current lack of a holistic approach are rather than just revisit several key papers. A much stronger critique and way forward would be very helpful.
- Response: In line with this suggestion we attempted to point out the need for more a widely-applicable management approaches for addressing challenges accruing from flash flooding events in the last section and emphasized on the opportunities for future research on the specific field on small business management.
<><><>
Minor comments (refers to line numbers):
46 - 'even their mere survival' is a bit dramatic.
- Response: The argument/phrase is moderated in the revised manuscript.
<><><>
59 - 'profound'? - correct term?
140-141 - first sentence seems redundant
164 - 'serious and strong'? - says who?, why?
Table 1 - reduce font size? Table 2012?
267 - remove 'actually'
293 - ()?
298 - messo / meso?
438 - 'excerpt' ? Table 2 - is this really necessary - not clear what it adds other than a list of pubs.
560/645/681/782 - font for the dod?
583 - weak reference - which jurisdiction etc?
- Response: These minor grammar/syntax/format mistakes and inconsistencies are now corrected in the revised manuscript.
<><><>
Table 1 etc. what is the linking lens between these theories and which school of philosophy do they all come from? This needs teasing out. What is the position held by the authors and how are they viewing the SME world?
- Response: We added the following text in the revised manuscript in order to respond to the above comment (revised manuscript, page 4-5):
“(…)All these theoretical frameworks have been utilized to disaggregate the different approaches and explain firms’ responses to the challenges arising from sustainable development under the scope of climate and weather-related hazards. A common ground for the development of these theories is that they recognize that the mere focus on financial goals is inadequate to guide firms to success. Environmental and social parameters should be integrated into corporate strategy in order for firms to thrive in a complex and turbulent environment”.
Likewise, in p.3 we denote that:
“(…) it is of critical importance to examine the wide spectrum of factors which facilitate or discourage SMEs to develop their resilience capacity due to the fact that the impacts of EWEs on SMEs could also bring significant problems at local, regional and/or national levels (for instance, supply chains experiencing long-term interruptions or ceasing to function). This is owing to the crucial role of the SMEs in the local societies as job providers and another explanation is that SMEs consist the vast majority of businesses operate both in developed and developing countries (Wedawatta et al., 2014; Ingirige and Wedawatta, 2011; Coates et al., 2020; Pathak and Ahmad, 2016; Marks and Thomolla, 2017; Samantha, 2018). Therefore, the great impacts of SMEs on the economic development, at all levels, clearly shows the necessity for effective tools for protecting them for EWES.
<><><>